# Qualitative Investigation into Therapists’ Experiences of Online Therapy: Implications for Working Clients

**DOI:** 10.3390/ijerph181910295

**Published:** 2021-09-29

**Authors:** Yasuhiro Kotera, Greta Kaluzeviciute, Christopher Lloyd, Ann-Marie Edwards, Akihiko Ozaki

**Affiliations:** 1College of Health, Psychology and Social Care, University of Derby, Derby DE22 1GB, UK; g.kaluzeviciute@derby.ac.uk (G.K.); c.lloyd@derby.ac.uk (C.L.); annm.edwards@icloud.com (A.-M.E.); 2Department of Psychiatry, School of Clinical Medicine, University of Cambridge, Cambridge CB2 8AH, UK; 3Department of Breast Surgery, Jyoban Hospital of Tokiwa Foundation, Iwaki 972-8322, Japan; aozaki-tky@umin.ac.jp; 4Medical Governance Research Institute, Tokyo 108-0074, Japan

**Keywords:** online therapy, COVID-19, qualitative, therapeutic relationship, thematic analysis, workplace mental health

## Abstract

Online therapy has increasingly been utilised during the COVID-19 pandemic by many, including working populations. However, few qualitative studies have explored how online therapy is experienced in practice and discussed its implications for those working clients. Semi-structured interviews attended by nine integrative psychotherapists practising in California, the United States, were conducted. Thematic analysis of the transcripts identified three themes: (i) ‘Positive experiences of online therapy’, (ii) ‘Challenges experienced by therapists and clients in online therapy’, and (iii) ‘Preparation and training for online therapy’. Online therapy was assessed as being helpful, particularly in terms of mitigating against previous geographical and temporal barriers to uptake. However, due to technological disruptions and potential blurring of professional boundaries, online therapy may detract from the emotional salience of therapy, negatively impacting the therapeutic relationship and containment. Considering these positive experiences, participants expected that the demand for online therapy would continue to increase. Particularly in the occupational context, online therapy can offer interventions without fostering shame regarding mental health. The findings provide preliminary qualitative evidence that online therapy can be a useful adjunct to traditional forms of face-to-face therapy. However, therapists require more explicit training in implementing online therapy. Results are discussed in particular regarding the utility of this therapy for working clients.

## 1. Introduction

### 1.1. Emergence of Online Therapy

In 2013, the American Psychological Association published ‘Guidelines for the Practice of Telepsychology’. Analogous to this, within the UK, the British Association for Counselling and Psychotherapy (BACP) recently published good practice guidance for therapeutic working online [1] Both developments underscore the rising interest in online therapy in the counselling and psychotherapy professions. They have paved the way for formal recognition of ‘telepsychology’, also known internationally as online therapy, e-therapy, e-counselling, computerised cognitive behavioural therapy (cCBT) and electronic cognitive behavioural therapy (eCBT) [2]. The term ‘online therapy’ (online therapy) in this paper predominantly refers to live video therapy. However, we also acknowledge that this term can include other online and integrative therapeutic support types, including asynchronous (email) and synchronous (instant messaging) communication, alongside live video therapy. Additionally, the terms ‘psychological therapist’ and ‘counsellor’ are used interchangeably and inclusively to refer to a range of trained practitioners who provide talking therapies and psychological intervention to individuals experiencing psychological and emotional distress.

Although online therapy is a relatively recent and emerging means of therapeutic working, using technology for therapeutic means is not new [3]. As early as the 1970’s, tape-recorded self-help approaches and computerised programs, which imitated person-centred therapists, were integrated into therapeutic approaches [4]. More recently, cCBT, a form of treatment during which clients receive pre-programmed responses based on CBT, has received significant scholarly attention, and has long been championed within the National Institute for Health and Care Excellence (NICE) guidelines for good practice, for both mild to moderate depression, and the treatment of phobias [5]. By contrast, however, online therapy, such as that provided through video or telephone technology, has received less attention in the literature, suggesting the need for further evaluation [6,7].

### 1.2. Strengths and Limitations of Online Therapy

At a broad level, findings from several large-scale meta-analyses have provided strong support for the adoption of online psychological interventions as legitimate standalone therapeutic interventions [8]. Indeed, Carlbring et al. [9] found that in-person versus eCBT were broadly equivalent in outcomes.

On a more idiographic and experiential level, however, the results are more mixed. Online therapy is generally recognised as having several benefits. One of the most important arguments in favour of online therapy is that it may assist in meeting a demand for psychological support that traditional face-to-face therapeutic interventions cannot always provide [10]. Second, online therapy may also offer individuals increased anonymity and privacy and may be offered at a reduced cost due to lower therapist overheads. These cost benefits may simultaneously open up access to previously disenfranchised and minority populations, who may have been excluded from therapeutic support due to economic status [11]. Beyond some of these pragmatic benefits, it has been suggested that online therapies may encourage more emotional expression and self-reflection [12]. However, the relational impact of these factors is more variable, and there are several challenging aspects to online therapy, which necessitate further exploration.

First, it is posited that online therapy may make it more challenging for therapists to identify and repair alliance ruptures [13] or cultivate a therapeutic presence with clients [14]. Here, one of the most obvious challenges is the lack of non-verbal or behavioural cues. Prior to the COVID-19 pandemic, [15] identified several important challenges to therapeutic relationships caused by the digital space and social media, including issues of therapist privacy (clients frequently search for therapist/counsellor personal information online, which inevitably impacts the therapeutic dynamic), virtual impingements (through which online discoveries alter the physical therapeutic relationship) and a desire to internalise digital versions of the therapist or digital communications (e.g., developing transference through email exchanges). Whilst these are all important aspects, it seems prudent to explore how therapists themselves have experienced the shift to online therapy during the COVID-19 pandemic.

### 1.3. Increased Demand for Online Therapy in Occupational Groups during the COVID-19 Pandemic

Among many populations that have benefitted from online therapy, the working population has particularly enjoyed greater benefits of online therapy [16]. While these findings illustrate clients’ experiences receiving online therapy, therapists’ perspective regarding online therapy during the COVID-19 pandemic remains to be evaluated. Such an appraisal will help to inform workforces to better understand how they can benefit from using online therapy. Accordingly, this study evaluates the perception regarding online therapy from a qualified therapist’s perspective. Specifically, online therapy allows individuals and groups to access psychotherapeutic services on demand. This is especially significant, as research indicates that nearly two-thirds of all people with diagnosable psychological disorders do not seek treatment [17,18]. Likewise, the low rate of help-seeking is salient among working populations, caused by various factors including shame regarding mental health problems, as identified in various working groups [19,20,21,22,23]. Online therapy is recommended, as a working client can access therapy more privately, bypassing their mental health shame; however, this has not been discussed in relation to therapists’ perspectives.

### 1.4. The Present Study

Despite theoretical and anecdotal consideration of the helpful and unhelpful aspects of online therapy, there is a general lack of in-depth qualitative exploration from the therapist’s perspective. Whilst some qualitative research has used broad surveys to explore therapists’ attitudes to online therapy during the COVID-19 pandemic [24], this has often not been carried out in an idiographic and inductive manner. Despite a growing body of research and the increasing normalisation of online therapies in the last decade, there is little qualitative research that attempts to explore how psychological therapists experience and make sense of their online therapeutic work with their clients. Hence, this paper aims to (a) appraise the perception towards online therapy from the therapist perspective (Aim 1) and offer suggestions for future therapeutic practice (Aim 2) through qualitative investigation, then (b) discuss how working clients can benefit from online therapy (Aim 3). Aim 3 was added post hoc, considering the ever-increasing demand of online therapy for this population, and to inform the readers for this Special Issue.

## 2. Materials and Methods

In this qualitative study, we sought to assess experiences of conducting online therapy during COVID-19 based on nine licensed and qualified therapists’ work conducted in California, the United States (US). Participants were recruited via social media through professional clinical networks. Online individual semi-structured interviews were the main data collection method, in addition to demographic questions presented to each research participant. Data were analysed using the thematic analysis method [25,26]. GK and YK held and transcribed the interviews; GK analysed the data and extracted the main themes emerging from the interviews; and all authors contributed to the discussion of the findings and further recommendations for online therapy practice. The study adhered to the Consolidated Criteria for Reporting Qualitative Studies (COREQ) guidelines [27]. All participants only knew the gender of the interviewers (GK or YK) before the interview. The details of the study method are explained below.

### 2.1. Study Design

Ethical approval for this study was granted by the University of Derby Research Ethics Committee 06-15-YK. The study used qualitative semi-structured interviews [28]. This method consists of a dialogue between the researcher and the participant, guided by a flexible interview schedule and supplemented by additional follow-up questions and probes [29]. The semi-structured interview method is appropriate for studies with as few as 8-12 participants, as it promotes the inclusion of multiple complex datapoints through iterative interactions between the interviewer and participant (and, as such, the goal is to qualitatively capture a complex phenomenon within its context rather than to measure an average parameter across a representative population, as in, for example, statistical studies) [30]. Experiential accounts of conducting online therapy before and during the COVID-19 pandemic are scarce. Our study sought to capture the potential strengths, limitations, and unique components of digital therapeutic exchanges (e.g., accessibility due to a lack of geographical constraints, online containment processes, flexibility with regard to time, experiencing therapy at one’s home location, and cost-effectiveness), which are experienced differently by each practising counsellor/therapist.

A pre-designed semi-structured interview schedule (see Appendix A) was developed and sent to all research participants in advance of the interview to provide some guidance. Our interview questions were guided by a similar study conducted by [31] on attitudes toward online therapy among therapist trainees in Turkey, which also used the semi-structured interview method. In addition to the questions used in Tanrikulu’s study, we developed additional questions idiographic in nature (e.g., concerned with the meaning of online therapy and its significance for specific patient populations/symptoms). Interviews were held online by using the MS Teams software established in the university system. All interviews were recorded and transcribed verbatim with the consent of the participants, who later confirmed the accuracy of the transcription. All participants were required to read the participant information sheet and sign a consent form. Participants were able to withdraw from the study at any time.

### 2.2. Recruitment

Purposive and snowball sampling techniques were used to recruit the participants, who practised in California, US. According to APA data [32], the State of California has the highest number of licensed psychologists in the country who are trained in various therapeutic modalities (in CBT, Gestalt, Transpersonal therapy, etc.). Nine licensed therapists/counsellors participated in an online interview. The details of the participants are reported in the results section.

### 2.3. Analytic Procedure

The study used thematic analysis to systematically identify and organise meaningful patterns across a dataset [26]. Since our study seeks to identify unique and/or divergent idiosyncratic experiences pertaining to online therapy, this method of analysis was deemed appropriate. Thematic analysis was carried out in the following order to identify the relevant themes: (i) Familiarisation, (ii) Generating initial codes, (iii) Searching for themes, (iv) Reviewing themes, and (v) Defining and naming themes [25] (each thematic analysis process is described below).

### 2.4. Reflexivity

It is important to acknowledge the role played by researchers’ ideas, thoughts, and feelings in thematic analysis [25]. Our study approached the research process from a critical realist standpoint [33]. Although critical realism acknowledges that our world is largely socially constructed (i.e., we cannot think about the world independently of our beliefs), it also nurtures the idea of developing realistic and causally meaningful interpretations for complex social phenomena. According to Outhwaite [34], one way to arrive at a realistic interpretation is immediately acknowledging the researcher’s vehicular social and epistemic role in the research process (reflexivity). Therefore, tracing how our social and linguistic practices influence and change research findings and analytic procedures is part of a critical realist analysis. In our study, a psychotherapy researcher (GK) who held some of the interviews coded the transcripts and developed master themes; the themes were then reviewed by a researcher in counselling and an accredited psychotherapist (YK), who also held some of the interviews, and a researcher and practising chartered psychologist in counselling psychology (CL), who was not involved in the interview process. This enabled a ‘cut and come again’ disposition [35], ensuring that no single causal account, theme or interpretation was accepted uncritically and that researchers were able to assess and compare contrasting research findings. All themes and data interpretations were checked and agreed upon by the researchers.

#### 2.4.1. Familiarisation

Interview data were read repeatedly to formulate initial interpretations, patterns, and themes [26]. Similarly, audio and video footage of interviews was viewed again to draw out initial thematic maps [25].

#### 2.4.2. Generating Initial Codes

The coding process in this study was ‘theory-driven’ [25], with a set of research questions that were identified before the interviews (Table 1) as well as focus areas identified through the interview schedule (online therapy, therapeutic relationship, the online medium, and client perspective) (Appendix A). This enabled a more comprehensive coding process. In total, 78 codes were identified from nine interview transcripts. Some of the example codes are included in Table 2 below.

#### 2.4.3. Searching for Themes

The previously identified codes (Table 1) were attached to theme-piles using Braun and Clarke’s mind map process in order to categorise the data at a broader level of analysis [25,26]. Specifically, codes across all four interview focus areas were compared in terms of similarity and overlap (code clusters). During this process, we identified the following themes (Table 2).

#### 2.4.4. Reviewing Themes

During this phase of the research, themes were analysed against the coded data as well as the entire dataset for coherency and relevance (details of which are available in the ‘Results’ section). Specifically, the identified themes (Table 2) were checked in relation to the study’s research questions [26]. The data were organised in the following manner: reports of good practice in online therapy, including the suitability of online therapy for specific clients and, in particular, clients in employment who may have time and/or geographical limitations which may otherwise pose barriers to receiving psychotherapy (corresponding to Theme 1, addressing research aims 1 and 3); challenging and problematic aspects of online therapy for both therapists and clients (corresponding to Theme 2, addressing research aim 1); and therapist experiences of training and guidelines for online therapy, including gaps in available information about online therapy ethics, digital platforms and suggestions for future training (corresponding to Theme 3, addressing research aim 2).

#### 2.4.5. Defining and Naming Themes

The collated data extracts were refined to ensure that each theme was consistent with the accompanying narrative [26]. Lastly, during the revision process, sub-themes were presented to enhance the clarity of our findings. Theme 1: ‘Positive therapist and client experiences of online therapy’ encompassed T1-1 ‘Beyond expectation’, T1-2 ‘Quality assurance’, 1-3 ‘Accessibility’, and 1-4 ‘Control over therapy’. Theme 2: ‘Challenges experienced by therapists and clients in online therapy’ contained T2-1 ‘Technological disruption’, T2-2 ‘Lack of containment’, T2-3 ‘Disruptive environment’ and T2-4 ‘Severe psychopathology’. Lastly, Theme 3: ‘Preparation and training for online therapy’ included T3-1 ‘Lack of training’, T3-2 ‘Lack of guidance’, T3-3 ‘Need for helpful online community’ and T3-4 ‘Need for evaluation’. Table 3 summarises the themes and sub-themes.

## 3. Results

The demographic information of the nine participating therapists/counsellors is as follows: seven females and two males, age M = 44.5, SD = 9.8 years, a high level of experience in both providing and receiving therapy (M = 14.2, SD = 6.6 years), and complex theoretical and clinical differences given the variety of therapeutic modalities practised by each participant (psychodynamic, humanist/existentialist, person-centred, gestalt, eclectic, CBT, attachment, etc.). Further demographic participant information is provided in Table 4.

### 3.1. Theme 1: Positive Experiences of Online Therapy

The majority of the participants reported that online therapy worked, or, at the very least, worked better than expected, leading to favourably shifting attitudes and openness toward online therapy post-pandemic (T1-1: Beyond expectation).

*Participant 8:* [Online therapy] was not part of my master’s program or my supervision. It’s something that I’ve learnt more about, now that I’m doing it since COVID. … and it works! And it’s better than I thought it was going to be.

*Participant 7*: Until August, I was thinking of online therapy as a poor substitute for in-person [therapy], and a limitation. … I think that there is a slight decrease in presence within therapy, and the quality of work I can do in this medium. Lately, in the last couple of months, I feel as I’ve settled more into the pandemic life in general, and also because I had some new clients come in and I’ve had some successes with them […] I’ve been considering more possibility of just maintaining an online practice for a couple of years, especially if I move. I notice myself being more open to it.

*Participant 5:* I see [online therapy] as an adequate substitution. It seems to be working okay, and in that sense, I feel really grateful that I can keep working.

In addition, some technological benefits were acknowledged for the quality assurance, including supervision as well as the development and sharing of therapeutic knowledge within therapists’ professional network and training (T1-2: Quality assurance).

*Participant 3*: I think it offers some additional benefits in terms of the digital technology, like the ability to record. That can be used for quality purposes, for supervisory purposes, to enhance the experience of the provider, knowledge. Some of the other technology-related benefits are being able to take notes, share the screen, show a video on the spot, certain things that we can always do by ourselves, but that can be a part of the shared space in therapy.

Participating therapists also reported positive aspects of online therapy from the client’s perspective, including geographical and temporal flexibility, increased access to therapy, increased number of providers (as well as the ability to choose a therapist based on their expertise/suitability rather than geographical proximity) and reduced costs (T1-3: Accessibility).

*Participant 8:* I think that online therapy does help with access to care because some people who, for example, don’t have childcare and they need help, but they can’t get away from home, they’re more easily able to access services or they’re able to find low-cost services. There seems to be more opportunity and availability online because they can be seen by providers from all over California instead of their town. So, I think there are some good pieces regarding access and equity and fairness.

*Participant 3*: The biggest benefits, I think, are the obvious ones: the flexibility that you have, the possibility of doing therapy from your own home, well, from both ends. It increases flexibility from the provider and the client. […] It’s shifting the landscape of therapy, and it’s coming on the heels of the movement where mental health is becoming a recognised field and an important aspect for the masses, not just the traditional psychoanalytic thinking in the Victorian times where patients were from higher socioeconomics. So, there’s that—the shifting in landscape where therapy is becoming more mundane, more accessible. Online therapy has accelerated it.

Online therapy is the only option for some clients due to geographical limitations (e.g., lack of available counsellors and therapists or time spent travelling) or other circumstances (such as illness or caretaking/parenting responsibilities). These aspects are particularly important for clients in employment who, in the past, did not have time or lacked geographical proximity to attend face-to-face therapy:

*Participant 7:* [My patient] was living across the bay from me, and she’s a mom also, so with traffic and parking and everything, it was taking her between 60 and 80 min to get here. So, for her it was really just like, “I can’t do this for therapy”. I think if she hasn’t had the online therapy, she would have probably stopped working with me.

Participants also offered some interesting reflections about having more control over the therapeutic situation and being less affected by potentially difficult and/or negative therapeutic experiences with clients (T1-4: Control over therapy).

*Participant 7*: If somebody’s got a lot of energy, you can shrink the window a little bit. You can turn down the volume if someone’s voice is very abrasive. And I actually am somebody who gave up on doing couples therapy, because I find that my system, my body cannot take the amount of stress and conflict that couples bring in. Like, I just want to shut down and run away screaming! So, I have actually recently thought about doing it online because I’m not feeling that tension with my body.

Overall, participating therapists evaluated online therapy more positively than expected (T1-1), and reported several advantages related to the quality assurance (e.g., use of recording for supervision; T1-2), increased accessibility from both therapist and client perspectives (T1-3), and having more control over therapy (e.g., adjustable size of screen and volume of sounds; T1-4).

### 3.2. Theme 2: Challenges Experienced by Therapists and Clients in Online Therapy

The most significant limitation of online therapy identified by all participants is limited physical contact and body language, both of which can be further diminished by technological disruptions (T2-1) (e.g., bad connection, poor video quality or lack of knowledge on how to operate a specific software or set up a camera). This was found to have a direct impact on the development of therapeutic relationships as well as containment processes (T2-2: Lack of containment).

*Participant 1:* When there are glitches with technology, it definitely affects the sense of containment. It is hard to rewind and get back to where a client had been, or what they had been expressing after a disruption—especially if they were crying.

*Participant 3:* [The fact] that it requires the technology itself, it’s not equally accessible by everyone who may not have the bandwidth for the required Internet speed, good Wi-Fi, computer, some knowledge of how to set up lighting around the cameras and some technical aspects that are, you know… it falls on the provider and the client, as opposed to coming to a room where things are set up and we all kind of know how it works.

*Participant 9:* I think that clients who have trauma [experiences] struggle more. They have a hard time to be present. Some people have more [expression] through their body language, so they need a more solid atmosphere.

*Participant 6:* [Therapy] feels less of a ritual… How to replace it? … I can’t control how the client comes into the session, how they’re sitting, their environment, distractions. For me there’s something sacred around the container and the preparation, and so I can both prepare and have my surroundings. I obviously have less control over the client [now]. I notice it a little bit more now and so I think it’s important to do what I can do from my end to hold that.

Some interesting observations were revealed by the participants about challenges caused by the blending of the home environment and the digital therapeutic space, which include disruptions from family members during therapy sessions, other technological interruptions (phones, laptops, tablets), lack of private space, and client behaviours that would not ordinarily occur during physical (face-to-face) sessions (T2-3: Disruptive environment).

*Participant 8:* With my teenage clients, I think that they don’t have as much respect for the therapy… [It’s] not the kids, but the parents will come into the room and say, “are you talking to [participant name]?”, and they’re like “yeah”, and then the [parents] will say something like, “ok, well, when you’re done, I’ll need you to do the laundry”, you know. … there’s more interruptions not only from their environment, their phones and whatever, or their cat or their baby, but also from other people living in the household who are reminding them of chores or whatever, so it’s harder to maintain focus.

*Participant 3:* I have noticed that clients from lower socioeconomic backgrounds tend to experience more distraction. I can give you some examples of what I mean: people who often do therapy in their cars are usually from poorer socioeconomics, which means less time, and they often quite literally don’t have the private space.

*Participant 7:* I had a client who… She will drink alcohol, she will have a cocktail or whatever, during the session… And that would just very rarely happen in your office. But if they’re home, and their fridge is right there, and it’s a time where they would kick back anyway, then this seems more natural to have a drink when you’re talking with a therapist. That was definitely surprising to me, to see that there’s a blending between someone’s home life and their therapy appointment. […] In that sense, the container-ontained relationship is out of control.

Several participants noted that online therapy is not suitable for clients who suffer from severe psychopathology or mental health distress (e.g., trauma or personality disorders) (T2-4: Severe psychopathology), because they require greater contact and containment that cannot be facilitated via online mediums:

*Participant 8*: I feel like right now I have a client who wants to transition to [face-to-face] psychotherapy because they are too severe, but because there’s this force to be online, I feel like one of the issues is that not everyone is well-suited for this mode of therapy.

In sum, the participants reported that technological issues (T2-1) could negatively impact the containment of the therapy (T2-2). Moreover, they noted that some clients were not in an ideal environment to engage in therapy (T2-3). Because of these challenges, they perceived online therapy not suitable to treat severe psychopathology (T2-4).

### 3.3. Theme 3: Preparation and Training for Online Therapy

All participants reported having only had minimal or no training for online therapy prior to the pandemic (T3-1: Lack of training).

*Participant 2:* I received zero training, even though I would have liked to receive education on online therapy. For example, what about privacy? Containment in the room?

Some of the issues identified in training for online therapy include a) lack of documents, surveys and scales for online sessions; b) lack of technological guidance (for both clients and therapists); and c) lack of guidance on how therapeutic relationships and outcomes can be addressed in online therapy (T3-2: Lack of guidance).

*Participant 8:* I’ve had to make certain documents online because they previously [didn’t exist] […] for example, certain anxiety, depression, relationship, satisfaction, mood surveys that I would typically do as a check-in just before the session with the clients to get the baseline of their functioning. The first two weeks of being online I couldn’t get them because there was no way to administer them. So, tools had to be developed specifically for online therapy.

*Participant 5:* I think that it would be really helpful just to understand what is exactly being expected from you and what are the differences between [online and face-to-face] therapy.

*Participant 3:* I still feel like there is a lack of more nuanced aspects that I was trained about in-person, for example, observing the space, the container of where you are with the client, the quality of the presence. We didn’t get too much into that when we did the training, and I think that’s generally very important. […] Similarly, I wish I have been taught the ground rules more from the beginning to asking about the address, you know, those tips like: make sure you check with the clients that they’re in a private space, that they do not have any distractions, even other screens, phones, things like that. It’s just not natural for people to do and I think they make a big difference, if you know this from the beginning and set it up. You avoid disruptions and the general loss of the quality.

In order to mitigate the lack of training and experience in online therapy, some participants joined an online community of therapists; however, more support is needed (T3-3: Need for a helpful online community).

*Participant 8:* When working online, I found it more difficult to do consults with other therapists because in real life you’re in an office and you can say, “hey, can I ask you a question about something?” or “do you have this resource?”. Fortunately, there are online consult groups that I am a part of, but there is that missing component of peer support. It’s more difficult online.

Lastly, some participants highlighted a need for a careful evaluation of online therapy while recognising the potential of this form of therapy (T3-4: Need for evaluation).

*Participant 3:* Online therapy is shaking up the field. Because a lot of people that could not access therapy can now access it. The rules of the language of therapy are changing, and it opens up the field for new interpretations. I do not think we know yet how digital technologies will change our consciousness, and how we manipulate it for therapeutic benefits. Much like we do not see how complexity of narrative is changed over digital technologies.

Taken together, the participants did not feel that they had had enough training (T3-1), nor that helpful guidance was available (T3-2). Though some of them accessed an online community of therapists, more support was needed (T3-3). Additionally, a need for empirical evaluation of online therapy was suggested (T3-4).

## 4. Discussion

Since the beginning of the COVID-19 pandemic, the use of online therapy has increased rapidly, and working populations have utilised this form of therapy and received its benefits. However, the existing research primarily focused on clients’ perspectives, missing an understanding of how therapists perceive and experience online therapy. Accordingly, we aimed to (i) examine the perception towards online therapy, (ii) offer suggestions for future practice, and (iii) discuss how employees can benefit from online therapy. Our analysis identified positive experiences (T1), challenges (T2), and preparation and training (T3) relating to online therapy (Aims 1 and 2). The participating therapists perceived online therapy positively, reporting more utility than expected (T1-1) relating to factors such as quality assurance (T1-2), accessibility (T1-3), and control over therapy (T1-4), while noting some challenges, including the technological disruption (T2-1), a lack of containment (T2-2), disruptive environment (T2-3), and unsuitability for severe mental illnesses (T2-4). A lack of training and guidance (T3-1, -2) was noted by the participants, indicating a need for a more helpful online community and the evaluation of online therapy (T3-3, -4) in the future. These findings are discussed below, regarding clients in employment (Aim 3).

One notable finding of our study is that, although the participating therapists felt that they had not been trained enough in online therapy (see T3), overall, they found it helpful and were willing to continue using it (see T1). In addition to their positive experience, they also reported the advantages of online therapy from the client’s perspective relating to time, location, and costs. These components contribute to the accessibility of online therapy, which may be particularly helpful to busy clients. During the pandemic, many employees were forced to work from home, yet online therapy offered access to treatment for these clients. Considering the increased rates of mental health problems in the workforce during the COVID-19 pandemic [36,37], the value of online therapy is high, suggesting a need for more robust education and preparation for this form of therapy. Specifically, guidance on the digital skills, intake assessments, and how therapeutic relationships and outcomes can be addressed was raised as an example for educational items. Indeed, many therapy regulatory bodies have produced information sources to educate their registered members about online therapy [38]; however, more research-informed guidelines need to be established.

Moreover, many therapists showed an intention to continue using online therapy, which can have implications for clients, including employed clients. For example, shame regarding mental health problems tends to be high in many occupational groups, reducing help-seeking in this population [5,8,39,40]. Online therapy can offer access to treatment for these shame-sensitive employees, as they can access therapy from home without any time and costs associated with physically accessing a therapy room. As mental health shame is strongly associated with poor mental health in many different occupational groups [21,39], access to therapy without causing shame can be a safer approach to protect employee mental health. Moreover, as many employees receive therapy, the normalisation effects may be present, reducing shame in order to facilitate help-seeking in the workplace [40]. This in turn can result in increased compassion in the organisation [41], which is linked to numerous advantages, such as collaboration, trust, and loyalty [42]. Longitudinal data are needed to evaluate the impact of online therapy in organisations.

While highlighting the positives of online therapy, challenges were also reported (see T2). Technological problems, including a lack of digital skills in therapists and/or clients, are among them. An unstable internet connection can disrupt the flow in a therapy session, negatively impacting the therapeutic relationship and outcomes. Moreover, while many clients can benefit from the flexibility of online therapy, some clients are not equipped with a good environment at home to focus on therapy (e.g., presence of other family members, including children). Alternative approaches for this population need to be considered. Therapeutically, the participating therapists noted the limited view of the client as a challenge; much information can be received from the physiology (e.g., posture, how they move their hands and feet, etc.), which is often excluded in online therapy. This type of information is particularly important when treating a client with severe mental health problems [43,44], and this is another area of challenge noted by the therapists. How appropriate online therapy is for severe mental health problems remains to be appraised, indicating a need for future research.

## 5. Study Limitations

There are several limitations arising from this research project, which are important to note. First, as with all small-scale qualitative projects, the data and findings contained within this project cannot be assumed to be representative of larger therapist groups. Indeed, this study was conducted in California, US, with a particular social and cultural representation of what online therapy is (or is not). More diverse and larger samples are needed in the future studies. Nevertheless, the themes arising from this study will be of interest to therapists and clients from a broad range of backgrounds and will give insight into ways of working therapeutically online. Second, as the therapists have been drawn from a wide and eclectic mix of therapeutic orientations, each with sometimes diverging conceptualisations of therapy, there is limited sample homogeneity, which has prevented the in-depth exploration of therapies in depth. Third, due to the nature of study recruitment, the sample was self-selecting: participants who took part likely had stronger views of online therapy. This has likely impacted the study results. To counter this, further studies which use a quantitative and larger-scale study design will be useful in exploring in further depth some of the initial themes generated from this study.

## 6. Conclusions

The demand for online therapy due to the COVID-19 pandemic is expected to continue to increase in the coming years. This study reported the first-hand experience of online therapy from the professional therapist’s perspective, regarding the advantages, challenges and workplace implications. While noting the small sample size, the findings will help (i) therapists refine their future practice, and (ii) working clients and workplace leaders to consider helpful applications of online therapy to improve individual and organisational mental health outcomes.

## Figures and Tables

**Table 1 ijerph-18-10295-t001:** Generating initial codes—example codes.

*Focus Area*	Initial Codes
*Online therapy*	Having a secure online therapy platform Willingness to conduct online therapy part-time post COVID-19 The use of online therapy for specific populations Clients with severe psychopathology are generally not suitable candidates for online therapy The introduction of online therapy as a way to break historical barriers in terms of physical distance, access and costs
*Therapeutic relationship*	Loss of body language Clients showing therapists items from their home environment during online sessions Frequent distractions from within the home environment (e.g., family disruptions) Knowing clients’ location in advance of the session would improve the online session Containment as the sacredness of the therapeutic space
*The online medium*	‘Way of being’ is lacking online The ‘goodness of fit’ between client and counsellor is more important online than face-to-face Difficulty in establishing a working alliance with a client online Preference for video communication Difficulties in picking up micro-emotions through video calls
*Client perspective*	Clients do not always have the privacy of a space suitable for online psychotherapy Online therapy helps with access to care Clients prefer online therapy due to easy access and temporal flexibility There remains a difference between client’s home space and the physical therapy space; the two do not always overlap

**Table 2 ijerph-18-10295-t002:** Themes, corresponding aims, and example comments.

No.	Theme (Corresponding Aim)	Example of Participant Comment
1	Positive therapist and client experiences of online therapy (1, 3)	[Online therapy] was not part of my Master’s program or my supervision. It’s something that I’ve learnt more about, now that I’m doing it since COVID. … and it works! And it’s better than I thought it was going to be (Participant 8).
2	Challenges experienced by therapists and clients in online therapy (1)	[The fact] that it requires the technology itself, it’s not equally accessible by everyone who may not have the bandwidth for the required Internet speed, good Wi-Fi, computer, some knowledge of how to set up lighting around the cameras and some technical aspects that are, you know… it falls on the provider and the client, as opposed to coming to a room where things are set up and we all kind of know how it works (Participant 3)
3	Preparation and training for online therapy (2)	There is something to the idea of clients feeling safe in their environment. So, if they’re already in a stressful environment and they’re logging in to talk to me, they’re still in that same environment so their body might be reacting in the same way. … so it’s hard to create an experience for clients in their own home (Participant 1)

Aim 1: Perception towards online therapy. Aim 2: Suggestions for future practice. Aim 3: How employees can benefit from online therapy.

**Table 3 ijerph-18-10295-t003:** Themes and sub-themes.

Themes	Sub-Themes
T1 Positive therapist and client experiences of online therapy	T1-1 Beyond expectation
	T1-2 Quality assurance
	T1-3 Accessibility
	T1-4 Control over therapy
T2 Challenges experienced by therapists and clients in online therapy	T2-1 Technological disruption
	T2-2 Lack of containment
	T2-3 Disruptive environment
	T2-4 Severe psychopathology
T3 Preparation and training for online therapy	T3-1 Lack of training
	T3-2 Lack of guidance
	T3-3 Need for helpful online community
	T3-4 Need for evaluation

**Table 4 ijerph-18-10295-t004:** Participant demographics.

Participant	Gender	Age	Years of Experience	Therapeutic Orientation	Target Symptoms	Target Population
1	Female	48	20	Psychodynamic	Anxiety, Depression	Adults
2	Female	44	7	Humanistic, Existential, Gestalt, Attachment	Anxiety	Adults
3	Male	39	11	Eclectic	Trauma, Psychosis, Bipolar symptoms, Substance abuse	Adults
4	Female	68	25	Psychodynamic, CBT, Eclectic	Depression, Anxiety, LGBT transition, ADHD	Couples, Adults, LGBT
5	Male	44	22	Existential, Humanistic, Gestalt	Grief, Anxiety, Anxiety related to gender identity	Bi-cultural, Queer youth
6	Female	41	15	Gestalt	Life transitions, Anxiety, Bereavement, Grief	Adults, Women
7	Female	44	9	Gestalt, Non-directive Play, Art Therapy, CBT	Anxiety, Depression, Neurosis	Asian-American Women, Adults, Children
8	Female	32	8	Psychodynamic, CBT	Transitional issues, Adjustment disorders, Depression	Women, Teens
9	Female	41	11	Humanistic, Experiential	Stress, Depression, Cultural issues, Anxiety	Adults

## Data Availability

The data presented in this study are available on request from the corresponding author. The data are not publicly available due to ethical restrictions.

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
