# Peer review of "Qualitative Investigation into Therapists’ Experiences of Online Therapy: Implications for Working Clients"

_ijerph, 2021, doi:10.3390/ijerph181910295_

Round 1

Reviewer 1 Report

The sample is very short and there no evidences that the conclusions of the interviews match the perceptions of each professional groups. On the other hand the data  of each interview is very sparse and vague.

Some patients real cases with anonymized data can be included in the research and increase the accuracy of the results. 

Author Response

Dear Reviewer 1,

Thank you for your helpful feedback. We have systematically revised our manuscript addressing the points you have raised. Please see our responses below. We hope this revised paper is now acceptable for publication. We extend our sincere gratitude to you for your feedback that has significantly helped to strengthen the paper.

Reviewer 1

Reviewer 1’s comment 1

The sample is very short and there no evidences that the conclusions of the interviews match the perceptions of each professional groups. On the other hand the data  of each interview is very sparse and vague.

Authors’ response 1-1

Thank you for your feedback. In line with your comment, clarity of relevant evidence and interview data has been provided (e.g., addition of sub-themes).

Reviewer 1’s comment 2

Some patients real cases with anonymized data can be included in the research and increase the accuracy of the results.

Authors’ response 1-2

In line with your comment, real case data is included in the manuscript, and the presentation of data is improved for clarity (methods section).

Reviewer 2 Report

Thank you for inviting me to review this qualitative analysis of the experience of therapists providing online therapy for working clients. This is a well written paper that clearly adheres to COREQ guidelines. My main concerns are in the quality of the analysis reported, namely that you only report two main themes in the text and do not attempt to take the analysis further and group these under sub themes. My other concern is how the data on working clients are presented.  It is in the title of the paper and is listed as an aim but it doesn’t appear to come into the interview schedule or the results section clearly.

Introduction

You introduce the current study on line 38 of the first page which isn’t typical for a research paper. You should aim to funnel from wider topics of focus down to the current study at the end of the introduction. I would recommend reordering your introduction so you start with section 1.2.

Aim 3 does not appear to be addressed in the interview schedule so I feel this needs better  justification as an aim as currently it appears somewhat post-hoc

METHODS

These details that you include in section 2.2 really belong in the results section. “(N=9; seven females, two males) from a wide age range (M=44.5, SD=9.8 years), with a high level of experience in both providing and receiving psychotherapy and counselling (M=14.2, SD=6.6 years)”. Table 1 should also be relocated to the results section.

RESULTS

I wonder if the authors have attempted to drill down a bit with their analysis and bring in sub-themes to their two main themes. As it is, it is just a presentation of pros and cons, without a deeper exploration of what is being presented under these. You bring this in to the text but the way it is currently presented isn’t clear.

Theme three mentioned in the abstract also appears to be missing from the text.

Minor point, but I think it would be helpful to indent the quotes you use in the results as it’s hard to read as it is currently formatted.

DISCUSSION

I have not reviewed the discussion in detail at the moment due to the changes required to the rest of the paper first.

Author Response

Dear Reviewer 2,

Thank you for your helpful feedback. We have systematically revised our manuscript addressing the points you have raised. Please see our responses below. We hope this revised paper is now acceptable for publication. We extend our sincere gratitude to you for your feedback that has significantly helped to strengthen the paper.

Reviewer 2’s comment 1

Thank you for inviting me to review this qualitative analysis of the experience of therapists providing online therapy for working clients. This is a well written paper that clearly adheres to COREQ guidelines. My main concerns are in the quality of the analysis reported, namely that you only report two main themes in the text and do not attempt to take the analysis further and group these under sub themes. My other concern is how the data on working clients are presented.  It is in the title of the paper and is listed as an aim but it doesn’t appear to come into the interview schedule or the results section clearly.

Authors’ response 2-1

Thank you for your feedback. In line with your comment, adjustments have been made. First, the title has been adjusted as the implications for working clients are not qualitatively examined, only discussed. The abstract also clarified this aspect now. Moreover, ‘1.3 The Present Study’ section has been adjusted, again to highlight that qualitative investigation was done for Aims 1 and 2, whereas Aim 3 was discussed.

To clarify the context, the following has been agreed with the editorial board when they invited us for this issue:  This study was originally designed to explore perceptions of online therapy from the therapists’ perspective. To adjust to this invited issue, we added implications for working clients.

Reviewer 2’s comment 2

Introduction

You introduce the current study on line 38 of the first page which isn’t typical for a research paper. You should aim to funnel from wider topics of focus down to the current study at the end of the introduction. I would recommend reordering your introduction so you start with section 1.2.

Authors’ response 2-2

Thank you for your helpful suggestion. The structure has been rearranged as suggested.

Reviewer 2’s comment 3

Aim 3 does not appear to be addressed in the interview schedule so I feel this needs better  justification as an aim as currently it appears somewhat post-hoc

Authors’ response 2-3

In line with your comment, a further justification is now added, considering the ever-increasing demand of online therapy for this population and the readers for this special issue (at the bottom of ‘The Present Study’).

Reviewer 2’s comment 4

METHODS

These details that you include in section 2.2 really belong in the results section. “(N=9; seven females, two males) from a wide age range (M=44.5, SD=9.8 years), with a high level of experience in both providing and receiving psychotherapy and counselling (M=14.2, SD=6.6 years)”. Table 1 should also be relocated to the results section.

Authors’ response 2-4

In line with your suggestion, the demographic information of the participants is now placed in the results section. Accordingly, the subheading of 2.2 is changed to ‘Recruitment’ only; the word ‘Participants’ is removed.

Reviewer 2’s comment 5

RESULTS

I wonder if the authors have attempted to drill down a bit with their analysis and bring in sub-themes to their two main themes. As it is, it is just a presentation of pros and cons, without a deeper exploration of what is being presented under these. You bring this in to the text but the way it is currently presented isn’t clear.

Theme three mentioned in the abstract also appears to be missing from the text.

Minor point, but I think it would be helpful to indent the quotes you use in the results as it’s hard to read as it is currently formatted.

Authors’ response 2-5

In line with your comment, now sub-themes are added to improve our presentation of the results, including Theme 3. Indentation has been implemented for quotes throughout.

Following those revisions, changes are made in the discussion section.

Round 2

Reviewer 1 Report

The sample is very small and there are no evidences of significance.

So the conclusions can not be generalized.

Please try to define better the sample and improve it.

Author Response

Response Letter

Manuscript ID: ijerph-1364436

"Qualitative Investigation into Therapists’ Experiences of Online Therapy: Implications for Working Clients”

Dear Reviewers 1,

Thank you for your helpful feedback. We have systematically revised our manuscript addressing the points you have raised. Please see our responses below. We hope this revised paper is now acceptable for publication. We extend our sincere gratitude to you for your feedback that has significantly helped to strengthen the paper.

Reviewer 1

Reviewer 1’s comment 1

The sample is very small and there are no evidences of significance. So the conclusions can not be generalized. Please try to define better the sample and improve it.

Authors’ response 1-1

Thank you for your helpful feedback. In line with your comment, the wording of the conclusion is now amended, and caution to interpret our findings due to the sample size is added. Changes are highlighted in red.